# Pathomechanism of Pruritus in Psoriasis and Atopic Dermatitis: Novel Approaches, Similarities and Differences

**DOI:** 10.3390/ijms241914734

**Published:** 2023-09-29

**Authors:** Agnieszka Kaczmarska, Dominika Kwiatkowska, Katarzyna Konstancja Skrzypek, Zbigniew Tadeusz Kowalewski, Kamila Jaworecka, Adam Reich

**Affiliations:** 1Department of Dermatology, Institute of Medical Sciences, Medical College of Rzeszow University, 35-055 Rzeszów, Poland; agnieszkakacz70@gmail.com (A.K.); dominika.kwiatkowska1808@gmail.com (D.K.); kamilajaworecka@gmail.com (K.J.); 2Department of Pathophysiology, Wroclaw Medical University, 50-367 Wrocław, Poland; kate.theviolinist@gmail.com (K.K.S.); zbigniew.kowalewski@student.umw.edu.pl (Z.T.K.)

**Keywords:** pathogenesis, pruritus, itch, psoriasis, atopic eczema

## Abstract

Pruritus is defined as an unpleasant sensation that elicits a desire to scratch. Nearly a third of the world’s population may suffer from pruritus during their lifetime. This symptom is widely observed in numerous inflammatory skin diseases—e.g., approximately 70–90% of patients with psoriasis and almost every patient with atopic dermatitis suffer from pruritus. Although the pathogenesis of atopic dermatitis and psoriasis is different, the complex intricacies between several biochemical mediators, enzymes, and pathways seem to play a crucial role in both conditions. Despite the high prevalence of pruritus in the general population, the pathogenesis of this symptom in various conditions remains elusive. This review aims to summarize current knowledge about the pathogenesis of pruritus in psoriasis and atopic dermatitis. Each molecule involved in the pruritic pathway would merit a separate chapter or even an entire book, however, in the current review we have concentrated on some reports which we found crucial in the understanding of pruritus. However, the pathomechanism of pruritus is an extremely complex and intricate process. Moreover, many of these signaling pathways are currently undergoing detailed analysis or are still unexplained. As a result, it is currently difficult to take an objective view of how far we have come in elucidating the pathogenesis of pruritus in the described diseases. Nevertheless, considerable progress has been made in recent years.

## 1. Introduction

Psoriasis (PSO) is a chronic, immune-mediated inflammatory skin disease with a prevalence of about 1–3% of the general population depending on the ethnicity [1]. PSO symptoms include erythematous patches with silvery scales, mostly located on the scalp, elbows, and knees. In the past, PSO was considered a limited- skin disease, but currently, it is known that PSO is associated with various comorbidities, including psoriatic arthritis, mental health conditions, and cardiometabolic diseases [2,3]. Despite its high prevalence, the detailed pathogenesis of PSO is still not fully elucidated [4,5]. However, the disease typically develops in genetically predisposed individuals, and its occurrence is determined by immunological and environmental factors [6,7]. Recently, a significant breakthrough in treatment has emerged, particularly in the realm of biologic therapy. According to the guidelines set forth by the American Academy of Dermatology-National Psoriasis Foundation, biologic medications are recommended as a viable first-line treatment option for moderate to severe plaque psoriasis [8,9,10]. Biologic drugs encompass various classes, including those that inhibit tumor necrosis factor α (TNF-α) such as etanercept, adalimumab, certolizumab, and infliximab [11,12,13]. Additionally, there are other biologic drugs targeting specific cytokines, such as IL-17 (secukinumab, bimekizumab, brodalumab, ixekizumab), the p40 subunit of interleukins IL-12 and IL-13 (ustekinumab), and the p19 subunit of IL-23 (e.g., guselkumab, risankizumab, tildrakizumab, mirikizumab) [14,15,16,17]. In addition to biologic therapies, there are oral treatment options available for psoriasis. These include traditional agents like methotrexate, acitretin, and cyclosporine, as well as advanced oral medications [18,19,20]. When psoriasis is localized to a small area, typically involving less than 3–5% of the body surface, the primary treatment options involve the use of topical agents. These agents may encompass a variety of treatments, such as corticosteroids, calcineurin inhibitors, vitamin D3 analogues, keratolytics, and combination topical agents. Additionally, for the management of plaque psoriasis, light therapy can be employed, with narrowband UV-B phototherapy being a commonly used approach [21,22,23].

Atopic dermatitis (AD, eczema) is another chronic, inflammatory skin disease commonly manifested by dryness, erythema, scaling, crusting, and lichenification. In highly developed countries, the prevalence of AD appears to have stabilized at around 10–20% [24], while in developing nations, the rates are lower, however, with a rising trend [25]. Approximately 60% of AD cases manifest during the first year of life, although the condition can develop at any age [26]. The sensation of itch, along with sleep deprivation and the embarrassment (caused by visible skin lesions) significantly impact the wellbeing of patients with AD [27]. It is also remarkable that many AD patients may also develop conditions like asthma, allergic rhinitis, or food allergies [28]. The standard treatment approach for AD primarily focuses on the application of topical anti-inflammatory preparations and maintaining skin moisturization [29]. Systemic treatment becomes necessary when the signs and symptoms of AD cannot be effectively controlled with topical treatment (emollients and moisturizers, corticosteroids, calcineurin inhibitors) or UV light therapy alone [30]. It is also beneficial for reducing the overall use of topical corticosteroids, especially in situations where patients require large quantities over extensive areas of the body for extended durations. Traditionally, systemic treatment options for challenging AD cases included relatively broad-spectrum immunosuppressive drugs like cyclosporine, azathioprine, mycophenolate mofetil, and methotrexate [31]. Recent advancements in our understanding of the immunopathology of atopic dermatitis have led to the identification of specific molecular targets for biologic therapy. These biologic treatments can be categorized based on their mechanisms of action, and they offer new avenues for managing AD. Some of the categories and corresponding therapies include IgE-targeted therapy (Omalizumab), anti-IL-4 therapy (Dupilumab), anti-IL-4/IL-13 therapy (Tralokinumab, Lebrikizumab), IL-31-targeted therapy (Nemolizumab), thymic lymphopoietin targeted therapy (Tezepelumab), JAK inhibitors (Tofacitinib, anti-IL-12/23 therapy (Ustekinumab), IL-22 blocker (Fezakinumab). Phosphodiesterase inhibitors (Crisaborole and Apremilast) are examples of phosphodiesterase inhibitors [32,33,34,35,36,37,38,39].

Both conditions (PSO and AD) demonstrate significant differences, both on the pathologic and clinical levels. However, they also share some similarities, including the presence of subjective symptoms, such as pruritus. Pruritus is defined as an unpleasant sensation that triggers the desire to scratch. Nearly a third of the world’s population may suffer from pruritus during their lifetime [40]. This symptom is mainly prevalent in various inflammatory skin diseases with approximately 70–90% of PSO patients and almost every patient with AD suffers from pruritus [41,42,43]. In the treatment of itching associated with PSO and AD, there are various topical therapeutic options available, each with varying degrees of effectiveness. Patients often find relief by using emollients or moisturizers, which help optimize the skin’s pH and may contain anti-itch ingredients such as corticosteroids, calcineurin inhibitors, capsaicin, or local analgesics [44]. Despite the differing pathogenesis of AD and PSO, the complex intricacies among numerous biochemical mediators, enzymes, and pathways seem to play a crucial role in both conditions [45,46].

Based on the pathogenesis, pruritus can be divided into four different clinical domains: pruritoceptive/dermatologic (caused by activation of pruriceptors directly in the skin), systemic (due to peripheral or central nervous system response to circulating pruritogens), neuropathic/neurologic (as a result of central or peripheral nervous system damage/activation), or psychogenic ones (occurring in the psyche, often caused/exacerbated by psychosocial problems or psychiatric diseases). However, more than one domain may be present in one itchy disease (so-called mixed pruritus) [47,48,49]. In the past, histamine was thought to be the ultimate mediator of pruritus (histaminergic pathway of itch), but over the last decades, many other mediators have been identified to cause itch (nonhistaminergic pathway of itch) [50]. It is believed that histamine may play a predominant role in acute pruritus, while chronic pruritus often emerges from the activation of a non-histaminergic itch pathway, potentially mediated by the immune system. Consequently, chronic pruritus, prevalent in various chronic inflammatory skin conditions may arise from prolonged and repeated immune activation, accompanied by a cytokine storm, and an increased release of various pruritogenic mediators [51]. Due to chronic inflammation, the skin thickens, the epidermis changes morphologically, and the infiltration of immune cells (in particular T lymphocytes) can be observed within the skin. Conversely, acute pruritus, such as a reaction to an insect bite, often involves epidermal damage, with keratinocytes and local immune cells releasing various chemical mediators in response to the injury [52,53]. Mast cells play an important role in this process by degranulating and secreting their cytosolic granules, which contain histamine, leukotrienes, serotonin, proteases, cytokines, and other mediators [54]. Histamine and other signals cause local vasodilation and attract circulating immune cells, including leukocytes and neutrophils, to the region of injury to remove the potential pathogen and repair the damage. Changes in the local chemical environment are detected by sensory nerve endings, leading to itching. As opposed to chronic pruritus, this reaction normally resolves within a day or two [55].

Despite the widespread prevalence of pruritus, the pathogenesis of this symptom in various conditions remains elusive. This review aims to summarize current knowledge about the pathogenesis of pruritus in PSO and AD.

## 2. Data Sources and Study Selection

This review was carried out using an electronic literature search of the PUBMED database. Index terms included combinations of consecutive words: “psoriasis” or “atopic dermatitis” paired with “pruritus”, “itch” or “itching”. Articles published to December 2022, were taken into consideration. Our search yielded a total of 7944 results containing the mentioned keywords when reviewing PUBMED. First, articles published in languages other than English were excluded (*n* = 6333). In the next step, the search was restricted to books and documents, clinical trials, meta-analyses, randomized controlled trials, reviews, and systematic reviews. Finally, 644 scientific articles focusing on the pathomechanism of pruritus in psoriasis and atopic dermatitis were included in the review. All selected articles were reviewed in detail for relevance, ultimately yielding 62 articles. Additionally, we conducted a reference search within the included studies to identify any additional relevant articles.

## 3. Results

### 3.1. Pruritus in Psoriasis

PSO is triggered by the activation of the immune system mediated by various cells and cytokines, including T helper cell type 1 (Th1), T helper cell type 17 (Th17), interleukin-17 (IL-17), interleukin-23 (IL-23) and tumor necrosis factor-alpha (TNF-α). This process leads to hyperproliferation and premature differentiation of keratinocytes, which results in the development of scaly plaques on the skin [56,57]. About 60–90% of PSO patients suffer from pruritus, often severe and causing significant impairment of patients’ quality of life [58,59]. Damiani et al. studied factors associated with the severity of psoriatic pruritus. In a group of 10,802 patients treated with PSO, 33.2% of patients experienced mild pruritus, 34.4% moderate pruritus, 18.7% severe pruritus, and 13.7% experienced very severe pruritus. Demographic characteristics associated with greater severity of pruritus were lower secondary and primary education compared to higher education and female gender, while age, previous remission of PSO, or drinking alcohol were not associated with pruritus severity [60]. A study by Hawro et al. attempted to characterize pruritus and sleep disturbances and how they impair the overall quality of life of 104 patients with PSO. Overall, more than 39% of patients reported sleep disturbances. Patients, who report sleep disturbances had a lower quality of life. As for the pruritus patients without pruritus had better overall quality of life, measured by the World Health Organization QoL-BREF, in all domains except the psychological and social domains compared to patients with moderate to severe pruritus. There was no difference in depressive mood and anxiety between patients with and without pruritus [61].

Untreated pruritus in psoriasis may easily lead to a vicious circle—extensive itching and scratching of the skin can lead to trauma and, due to Koebner’s phenomenon, lead to the development of new psoriatic lesions, that again are itchy [62]. In recent years, biological treatments have significantly improved the outcome of PSO, however, treatment options for pruritus in PSO are still limited, although biologics seem to be promising antipruritic agents [63].

#### 3.1.1. Chemokines

Chemokines are small proteins that regulate the immune system by signaling through chemokine receptors to induce e.g., immune cell migration, mobility, and infiltration into the tissue [64]. Their importance in PSO involves the recruitment and activation of T lymphocytes, neutrophils, and macrophages [65].

A study conducted by Purzycka-Bohdan et al. analyzed the relationship between serum chemokine levels with disease and pruritus severity. Among the studied chemokines (C-C Motif Chemokine Ligand (CCL) 2/MCP-1, CCL3/MIP-1α, CCL4/MIP-1β, CCL5/RANTES, CXCL8/IL-8, CCL17/TARC, CCL18/PARC, CCL22/MDC), only CCL17/TARC showed a positive correlation with pruritus intensity [66]. It has also been reported that CCL17/TARC affects pruritus severity in AD patients [67]. Possibly, CCL17/TARC levels could be used as a biomarker of pruritus severity, but studies on a larger group of patients are needed [68].

The role of chemokines was also highlighted in the study conducted by Nattkemper et al. who aimed to identify mediators and receptors associated with pruritus based on RNA sequencing. This technique allowed precise identification of genes and quantification of their expression, even when genes are expressed at very low levels. The authors examined paired biopsies (non-itchy, non-lesional skin versus itchy, lesional skin) from 25 patients with AD, 25 patients with PSO, and 30 healthy controls. It was shown that the expression of genes encoding inflammatory mediators such as CCL4, CCL7, CCL8, CCL20, interleukin-19 (IL-19), IL-20, IL-26, IL-36A, IL-36G, and TNF-α) was unique to psoriatic skin with pruritus [69].

#### 3.1.2. Interleukins

In recent years, more and more attention has also been directed toward understanding the role of interleukin-31 (IL-31) in pruritus [70]. IL-31 is involved in inflammation, impaired barrier function, and pruritus [71]. Epidermal keratinocytes are the target of IL-31 and it was shown that IL-31 can cause the proliferation of basal cells. The change in barrier function results at least partly from decreasing the expression of filaggrin protein [72]. Elevated levels of IL-31 have been observed in pruritic skin diseases, such as prurigo nodularis, AD, bullous pemphigoid, cutaneous T-cell lymphoma, and PSO [73,74,75].

There are hypotheses that IL-31 is associated with promoting the growth of sensory neurons and their stimulation, resulting in increased sensitivity to pruritus [76,77]. Studies in mice have indicated that a single intradermal injection of IL-31 induces a strong scratching response lasting about 24 h, while continuous intradermal administration of IL-31 increases epidermal basal cell proliferation and thickening of the epidermal layer [78,79]. Of note, there is 31% amino acid homology between mouse and human IL-31. The genes for this cytokine are located on chromosome 5 and chromosome 12q24.31 [80]. Although IL-31 appears to contribute to the induction of pruritus in PSO, its significance regarding its effect on pruritus intensity in PSO is divergent [81]. However, both IL-31 and its receptor seem to be a potential therapeutic target for pruritic diseases, although there are some inconsistencies in the literature about the effect of its concentration on the severity and exacerbation of psoriatic pruritus [82]. Additionally, in a mouse study by Nocchi et al., activation via near-infrared illumination of a phototoxic agent that selectively targets itch-sensing cells, specifically through pyrogenic IL-31, demonstrated a reduction in itch-related behavior [83]. Purzycka-Bohdan et al. examined 300 adults treated for PSO and 186 control group individuals to investigate IL-31 gene polymorphisms and IL-31 serum concentration. Pruritus was observed in 97.4% of the PSO patient group, with a mean severity measured by the Visual Analogue Scale (VAS) of above six points. IL-31 serum levels were substantially elevated in PSO patients vs. healthy controls. However, serum IL-31 levels did not correspond with PSO severity or pruritus intensity (Figure 1) [66].

Narbutt et al. studied the effect of narrow-band ultraviolet (NB-UVB) radiation treatment on the serum levels of IL-31 and various neuropeptides in PSO patients. In this group of 59 patients, 80% of individuals suffered from psoriatic pruritus. The mean pruritus score according to VAS at the beginning of the study was 7 points and decreased after 10 irradiations to 2.4 points and 1.05 points at the end of therapy (20 irradiations). IL-31 concentration was also significantly decreased from 748.6 ng/mL at baseline to 631.7 ng/mL at the end of NB-UVB therapy [84]. The reduction in IL-31 levels during irradiation appears to correspond with a decrease in pruritus, suggesting a significant role of IL-31 in the pathogenesis of psoriatic pruritus. Interestingly, Bodoor et al. while studying the effects of IL-4, IL-13, IL-31, and IL-33 on inflammatory processes and pruritus in patients with PSO revealed that patients with PSO had significantly increased levels of these cytokines compared to other patients. Surprisingly, serum levels of these interleukins were not correlated with the severity of pruritus or the disease itself [85].

The involvement of Th17 cells and, more directly, IL-17 in human plaque PSO has been repeatedly confirmed (Figure 1) [86,87]. Ixekizumab, a high-affinity monoclonal antibody that selectively targets IL-17A, showed significantly better improvement of itch severity upon treatment compared to placebo or etanercept in moderate-to-severe PSO patients. Interestingly, there are hypotheses that this is not a direct effect on reducing pruritus, but the drug works indirectly by suppressing the inflammatory effect of IL-17 [88].

#### 3.1.3. Substance P and Neurokinin 1 Receptor

Neuropeptides are proteins secreted from nerve endings, although some other cells may synthesize them as well. Substance P (SP) is a neuropeptide involved in afferent neuronal signal transduction and is released after activation of sensory neurons in the skin [89,90]. SP acts via neurokinin 1 receptor (NK_1_R) also known as tachykinin receptor 1 (TACR-1) or SP receptor. NK_1_Rs are located on various tissues and cells, including skin, smooth muscle, immune cells, endocrine and exocrine glands, neurons in the central and peripheral nervous systems, and tumor cells [91,92]. NK_1_R on keratinocytes and fibroblasts, upon activation with SP, stimulates the secretion of interferon γ (IFN-γ), IL-8, and IL-1β, while on mast cells leads to degranulation and release of histamine, prostaglandin D2, leukotriene B4, vascular endothelial growth factor and TNF-α [93]. Clinically, the release of these mediators modulates the inflammatory processes and causes vasodilation, erythema, edema, and pruritus of the skin [94].

Twenty years have passed since the discovery that in the lesional skin of patients with PSO and pruritus there is an elevated level of SP in contrast to those with PSO without pruritus [95]. However, the role of SP in PSO and pruritus remains controversial. Reich et al., evaluating the effect of neuropeptides on itching in PSO patients observed weak, but significantly negative, correlations between pruritus intensity and SP plasma level [96]. However, Kongthong et al. studied SP levels in PSO of patients with pruritus before and after NB-UVB therapy. It was shown that prior to irradiation the levels of truncated SP were significantly higher in participants with PSO compared to those without PSO. Interestingly, after 20 irradiations, SP significantly decreased in the experimental group compared to patients who were not irradiated, but despite a significant decrease in the Psoriasis Area and Severity Index (PASI) scoring, pruritus did not improve after irradiation. Kongthong et al. suspected that pruritus after irradiation was present due to skin dryness as a side effect of therapy, rather than because of PSO [97]. To study the cutaneous response to SP in pruritic and non-pruritic areas in PSO patients and healthy controls, Amatya et al. intradermally injected SP, saline, and histamine. SP induced pruritus, exacerbation, and wheals in PSO patients, but these reactions were not different from those observed in healthy individuals [98].

#### 3.1.4. CD26/Dipeptidyl Peptidase IV Enzyme Activity

It is known that the CD26 molecule and serum dipeptidyl peptidase IV (DPP IV) activity are inversely correlated with disease activity in patients with systemic lupus erythematosus and may play a role in the pathophysiology of hepatocellular carcinoma, colorectal cancer, prostate cancer, and malignant mesothelioma [99,100,101,102]. However, DPP IV may also be involved in pruritus in PSO.

In an experimental study, Komiya et al. evaluated a potential link between CD26 with DPP IV enzyme activity and an increased risk of pruritus in PSO, using serum samples from PSO patients and in vivo experimental models of pruritus. The levels of DPP IV enzyme activity in the serum of PSO patients were markedly increased when compared to the healthy control group. The shortened form of SP digested by DPP IV was also significantly elevated in PSO patients and in an in vivo pruritus model induced by the full-length SP, scratching was reduced by treatment with a DPP IV inhibitor [103].

Kongthong et al. examined DPP IV levels in patients with PSO and pruritus before and after NB-UVB therapy. Serum DPP IV levels were lower in participants with PSO than in the group without PSO and NB-UVB treatment significantly increased the activity of DPP IV. However, despite its undoubtedly beneficial impact of NB-UVB on PSO lesions, no correlations were found between itch severity scale (ISS), VAS, and DPP IV levels [87]. Data on targeting CD26/DPP IV appear promising, but further research on this molecule is needed to better understand its role in pruritus in PSO and to contribute to the development of more effective antipruritic therapies [104].

#### 3.1.5. Transient Receptor Potential (TRP) Channels

Cutaneous sensory nerves (nociceptors) are one of the recently suggested therapeutic targets in PSO [105]. These nerves include Aδ and C fibers, which release neuropeptides; calcitonin gene-related peptide (CGRP) and SP are best known to be released upon activation of TRP vanilloid 1 (TRPV1) and transient receptor potential (TRP) channels. TRP channel activation has been shown to mediate neurogenic dermatitis, pain associated with arthritis, and pruritus [106]. Chemical and surgical sensory denervation directed at these nociceptors improves PSO in humans and mouse models, but the mechanisms are unclear [107,108].

Ozcan et al. examined the expression profiles of TRP channels in patients with PSO. They found reduced expression levels of TRPM4, TRPM7, TRPV3, TRPV4, and TRPC6 genes in PSO patients while the expression levels of TRPM2 and TRPV1 genes were elevated compared to the control group [109]. However, in another study on a group of patients with PSO and pruritus, increased expression of TRPV3 was only related to pruritus [69]. The importance of TRPV3 was already highlighted in postburn pruritus, AD, and allergic pruritus [110,111,112]. Experimental research attempts to attenuate itching through the TRPV3 inhibitors [113]. There are no studies on their use in psoriatic pruritus, but substances such as isochlorogenic acid isomers, *Tribulus terrestris*, forsythoside B, verbascoside, scutellarein, 17R-RvD, citrusinine-II have been used with success in AD or dermatitis [113,114,115,116,117,118,119,120,121].

#### 3.1.6. Nerve Growth Factor

Nerve growth factor (NGF) and its receptors stimulate the proliferation and survival of peripheral sensory and sympathetic neurons, increase the number of neurons in the brain, and stimulate angiogenesis, and T-cell activation [122]. Upregulation of NGF leads to hypersensitization of peripheral sensory nerves to pruritic stimuli [123]. An NGF landscape mapping study revealed that NGF activates the TRPV1 by binding to its receptor—tyrosine kinase A (TrkA) [42]. The dynamic neuroimmune network is regulated by the NGF-mediated TRPV1 signaling pathway, which plays a role in the regulation of neurogenic pruritus [124]. NGF regulates the synthesis of SP and CGRP, leading to an increase in their production. According to Yamaguchi et al. [125], elevated NGF and TrkA expressions are observed in PSO individuals who have pruritus, distinguishing them from PSO patients who do not experience itching. Nakamura et al. also reported increased NGF presence in lesional psoriatic skin with pruritus in comparison to non-pruritic skin and the expression levels of this protein correlated positively with the severity of pruritus [95].

#### 3.1.7. Opioid Receptors

Opioid receptors play a significant role in regulating itch sensation within the central nervous system [126]. In the brain, μ-opioid receptors (OPRM) have been associated with the intensification of itch, whereas activation of κ-opioid receptors (OPRK) may diminish or relieve pruritus. However, the specific involvement of opioid receptors in pruritus perception within the skin is not well comprehended [127].

In a study conducted by Kupczyk et al., who investigated the expression of opioid receptors in psoriatic skin, statistically significant differences in the OPRK system were found regarding itch. Specifically, the expression of OPKR in psoriatic plaques that were accompanied by itch was found significantly reduced compared to psoriatic plaques without itch. These findings suggest that the OPRK system may directly modulate itch perception in psoriatic skin, not limited to the central nervous system. The expression of OPRM did not exhibit any significant association with the clinical parameters examined, including PSO severity or itch intensity, as measured by the PASI and VAS, respectively. This suggests that OPRM expression may not play a prominent role in these specific clinical aspects across PSO patients [128].

In a study conducted by Taneda et al., it was found that the levels of OPRM and its endogenous ligand—β-endorphin, in the epidermis were similar between healthy individuals and psoriatic patients, regardless of whether they experienced itch or not. However, the levels of OPRK and dynorphin A were significantly reduced in the epidermis of psoriatic patients with itch compared to the levels observed in healthy controls. These findings suggest potential alterations in the OPRK system and dynorphin A signaling specifically related to itch in psoriatic patients [71].

Furthermore, the study conducted by Biglardi et al. involved the use of topical 1% naltrexone cream on the skin of patients with pruritus, which resulted in a marked anti-pruritic effect. This finding supports the notion that highly specific opioid ligands, which are unable to penetrate the blood-brain barrier, could be a promising therapeutic strategy for treating pruritus [129]. However, in light of the limited amount of research on the use of opioid antagonists, there is an urgent need for more research in this area to explore the potential benefits and therapeutic relevance of these substances in PSO and pruritus.

#### 3.1.8. Histamine

Histamine is one of the best-known pruritic mediators, however, nowadays, its role becomes a bit controversial, especially in the pathogenesis of itch in PSO. Some researchers have reported results that challenge the conventional view, pointing out that there is no significant correlation between pruritus intensity and plasma histamine levels. Moreover, there were no observable differences in plasma histamine levels between patients with and without pruritus [130]. One of the first findings, which suggested alternation in mast cell and histamine in psoriasis skin was a study conducted by Petreson et al. The study showed that people with psoriasis had higher levels of histamine in the skin and increased histamine release from mast cells compared to healthy subjects. During 6 months of ranitidine treatment, who is a competitive, reversible inhibitor of the action of histamine at the histamine H2 receptors, resulted in a reduction in PASI score from an initial 15.4 decreased to 5.8 [131]. Nakamura et al. observed an increased number of mast cells in psoriatic skin with pruritus. Moreover, they point out in their study that our current knowledge does not allow definitive confirmation of the role of histamine and/or SP in initiating pruritus in psoriasis. Nakumara et al. in their study also emphasized that in order to establish the specificity of histamine and/or SP involvement in PSO pruritus, a comparative study involving a large group of PSO patients with and without pruritus is needed [95]. In addition, the above-mentioned studies suggest that histamine may be locally overproduced in the dermis of PSO-affected skin, and that plasma histamine levels may not accurately reflect histamine levels in the skin itself. To provide further support for this concept, Domagała et al. conducted a study that examined the efficacy of two different histamine 1 receptor (H1R) antagonists: clemastine and levocetirizine as supplementary treatment to standard PSO therapy in reducing pruritus. The results showed a statistically significant reduction in mean VAS scale scores in the group receiving clemastine, and also in the group receiving levocetirizine. Regarding the PASI scores, the initial scores were comparable across all subgroups in the study. However, after the treatment, there was a significant reduction all patients, with the most notable decrease observed in the clemastine group. Nevertheless, the limitations of the study such as a short follow-up period and the small number of observed patients (*n* = 61) should also be considered [132]. Altogether, other histamine receptor subtypes should not be overlooked. Mommert et al. discovered that activation of the H4 receptor, which is prominently expressed on plasmacytoid dendritic cells (pDC) in PSO, results in an upregulation of IL-17 production, a cytokine that plays a critical role in the pathogenesis of PSO, contributing to the development and progression of the disease. These findings suggest that the H4 receptor and its association with pDCs may be an important target for potential therapeutic interventions in PSO [133]. Recently, it has been also shown that the blockade of H4R may help to inhibit spontaneous scratching behavior in mice [134].

### 3.2. Pruritus in Atopic Dermatitis

The pathophysiology of AD includes genetic, environmental, and psychological factors. AD arises from a complex interaction involving impaired skin barrier function, abnormalities in the skin microbiome, and dysregulation of the Th2 lymphocyte immune response, ultimately leading to disruption of the normal skin barrier and changes in skin thickness [135].

The immune response in AD can be divided into two phases. The first phase, the acute phase, shows an increased Th2-dependent immune response, including e.g., IL-4, IL-13, and eosinophils. In the second phase, the chronic phase, there is a predominant Th1/Th0 contribution, with substances such as IL-5, IL-12, IFN-γ, and granulocyte-macrophage colony-stimulating factor (GM-CSF) playing a significant role [50]. There are factors responsible for pruritus in both phases, however, it is still not elucidated which is the most important one and how their occurrence interacts with each other [136]. It is also known that external factors, such as the weather and climate in which the patient lives, as well as excessive sweating, can contribute to the onset or severity of itching [137,138].

#### 3.2.1. Thymic Stromal Lymphopoietin (TSLP) 

The thymic stromal lymphopoietin (TSLP) is highly expressed in cutaneous epithelial cells in AD and is even further released after mechanical trauma such as scratching [139]. Numerous studies have demonstrated that TSLP derived from epithelial cells stimulates T cells, dendritic cells, and mast cells, but the involvement of sensory neurons in this pathway remains less explored. Wilson et al. conducted a study to evaluate the mechanisms controlling TSLP secretion and the induction of TSLP-evoked itch. Their findings indicate that TSLP produced by keratinocytes directly activates sensory neurons, leading to itch behavior. The researchers also identified a specific subset of sensory neurons that require functional TSLP receptors and the ion channel TRPA1 to facilitate TSLP-evoked itch. Moreover, they revealed the ORAI1/NFAT signaling pathway to be a critical regulator of protease-activated receptor 2 (PAR2)-mediated TSLP secretion by epithelial cells [50]. Surprisingly, the use of tezepelumab—a human monoclonal antibody against TSLP—among 113 patients in a clinical trial showed limited efficacy regarding inflammatory symptoms and pruritus. Furthermore, subcutaneous tezepelumab (280 mg every 2 weeks plus topical corticosteroid) did not demonstrate statistically significant improvement in patients with moderate to severe AD vs. placebo plus topical corticosteroids at week 12 [36].

#### 3.2.2. TRP Cation Channels

As previously mentioned, TRP cation channels and G protein-coupled receptors (GPCRs) play crucial roles in inflammatory skin diseases and pruritus. However, both in PSO and AD, the exact role of the GPCR-TRP signaling pathway in itch transmission remains inadequately understood [110].

To explore the involvement of TRPA1 in itch transmission, Oh et al. used a specific TRPA1 antagonist, which selectively reduced itch-evoked scratching in mice. Furthermore, when mast cells were genetically deleted in the AD mice, itch-scratching behavior was significantly decreased and TRPA1 expression in the dermal, neuropeptide-containing nerve fibers was reduced [140].

In murine models of AD, the application of a topical TRPV1 channel inhibitor has been found to decrease scratching behavior induced by TRPV1 agonists. Furthermore, this compound was also shown to attenuate pruritus mediated by histamine and PAR-2. However, there was no improvement in pruritus mediated by serotonin or thromboxane. In addition to its effects on pruritus, further experiments indicated that the topical TRPV1 channel inhibitor may have a positive impact on barrier function in AD. In particular, murine treated with an oral TRPV1 channel inhibitor showed improvements in epidermal lipid recovery and a reduction in transepidermal water loss [141].

In the study by Wang et al., the potential protective effect of scutellarein, a compound found in several medicinal herbs known for its anti-inflammatory and anti-proliferative properties, was confirmed in mouse model of AD. The study identified TRPV3 as the molecular target of scutellarein. Moreover, through subcutaneous injection of scutellarein, the researchers observed significant suppression of pruritus, epidermal hyperplasia, and skin inflammatory response. These observations may suggest that scutellarein could be a potential therapeutic agent for managing AD and its associated symptoms. Despite promising results in animal models involving TRP cation channel inhibitors, data from clinical settings is still lacking [119].

#### 3.2.3. Histidine Decarboxylase

Histidine decarboxylase (HDC) is an enzyme that catalyzes the synthesis of histamine through decarboxylation of L-histidine in various cells, e.g., mast cells or basophils [142]. A couple of studies on various animals indicated that increased HDC expression in the epidermis takes part in eliciting pruritus [143].

HDC may also be important in AD. Inami et al. analyzing 12 patients with AD and 8 healthy subjects showed a positive correlation between epidermal HDC expression, which was significantly increased in AD patients, and VAS. It was noticed that the main cell types expressing HDC were keratinocytes, Langerhans cells, and melanocytes. The comparable expression level of HDC in melanocytes and a similar number of melanocytes in patients with AD and healthy subjects and the lack of correlation between HDC expression level in Langerhans cells and VAS suggest that those cells are less involved in AD-related pruritus. Thus, the correlation between HDC expression in the epidermis and VAS in AD patients is majorly caused by keratinocytes, which showed significantly increased HDC levels [144].

In line with these observations are the findings by Gutowska-Owsiak et al. They examined the expression of HDC in keratinocytes from AD patients and observed that keratinocytes express HDC protein extensively, and its level increases in atopic skin. Furthermore, exposure to factors like house dust mites, lipopolysaccharides, and certain cytokines that are associated with allergic inflammation led to increased HDC expression and upregulation of histamine levels in keratinocytes [145].

#### 3.2.4. Role of the Microbiota

The role of the microbiota in the modulation of itch is an essential aspect to consider in the context of AD. Children with AD had a significantly less diverse gut microbiome compared to healthy subjects [146]. In addition, studies of the skin microbiome also showed reduced microbiome diversity, especially regarding *Staphylococcus* sp., *Streptococcus* sp., and *Propionibacterium*. Increased *Staphylococcus aureus* (*S. aureus*) skin colonization has been observed, especially during AD exacerbations, and there is a correlation between the diversity of the skin microbiome and disease severity [147]. Interestingly, in a study conducted by Blicharz et al., it was found that skin colonization by *S. aureus* may be one of the factors exacerbating pruritus in AD [148]. Bacterial products, including lipoprotein of the cell wall known as lipoteichoic acid (LTA), might play a role in skin inflammation caused by staphylococci and can act as an agonist for Toll-like receptor 2 (TLR2). Since LTA had not been quantified in vivo previously, Travers et al. aimed to measure the amount of LTA and cytokines present in clinically infected skin lesions in children and compare it with the degree of skin inflammation. The results indicate that the quantity of LTA correlates with the patient’s Eczema Area and Severity Index score, suggesting that LTA could be an important component contributing to Staphylococcus aureus’ ability to exacerbate AD symptoms [149].

#### 3.2.5. Artemin 

Another mediator of pruritus in AD may be artemin which belongs to the glial cell line-derived neurotrophic factor (GDNF). In individuals with AD, there is an accumulation of fibroblasts that produce artemin within the skin lesions [150]. In vitro experiments conducted by Murota et al. showed that artemin prompted cell proliferation in a neuroblastoma cell line while injecting artemin into the skin of mice led to the sprouting of peripheral nerves and increased sensitivity to thermal pain. Mice treated with artemin exhibited scratching behavior when exposed to a warm environment, but those lacking the GDNF family receptor α3, a crucial receptor for artemin, did not display this behavior [151].

#### 3.2.6. Mas-Related G-Protein Coupled Receptors (MRGPR)

During inflammatory reactions in the skin, one of the skin reactions involves the release of antimicrobial substances. These substances include peptides such as cathelicidin, β-defensins, and the major basic protein of eosinophils. It is noteworthy that cationic antimicrobial peptides, such as β-defensins, and cathelicidin can induce mast cell degranulation through MRGPR [152]. When MRGPRs on mast cells are activated, this can lead to allergic reactions that do not involve IgE. These reactions can manifest as symptoms such as extravasation, pain, or pruritus [153].

In human skin diseases, researchers have observed GPR15L expression in inflammatory keratinocytes. The release of GPR15L can activate several MRGPRs present in sensory neurons and mast cells, ultimately leading to the development of pruritus and inflammation [154].

Moreover, previous studies have suggested that GPR15L may be a gene of interest associated with psoriasis [155]. Tseng et al. tried to identify potential genes responsible for mediating pruritus, we conducted a survey of publicly available raw RNA sequencing data. These data were derived from skin biopsies taken from patients diagnosed with PSO or AD, as well as from healthy individuals. The results of the study reinforce the idea that GPR15L is one of the genes that experience the most significant upregulation in diseased skin. Furthermore, our analysis revealed an increase in GPR15L expression in AD lesions compared to control samples, and this increased expression level appeared to be associated with disease severity [156].

#### 3.2.7. Interleukins

It is well known that the common α subunit of interleukin-4 (IL-4) receptors plays a significant role in the development of AD by blocking the signaling of type 2 cytokines such as IL-4 and IL-13. In addition, another key cytokine contributing to itch in AD is IL-31. The study by Nyggard et al. examined the levels of IL-31, and IL-33 with disease severity in AD patients and healthy controls. In AD patients, both IL-31 and IL-33 serum levels were higher in children than in adults. However, no correlation was observed between disease severity and serum levels of interleukins [157].

These cytokines involved in AD transmit signals through the Janus kinases (JAK) transduction pathways, with JAK1 signaling being a common component. For this reason, inhibition of JAK signaling appears to be a logical therapeutic approach for treating AD [158]. Recently, the European Medicines Agency has approved for the use of upadacitinib, an oral medication that inhibits JAK, in the treatment of moderate-to-severe AD. Napolitano et al. attempted to evaluate the efficacy of upadacitinib specifically for pruritus, upadacitinib in AD patients in real-life experience. Itch intensity was measured by Pruritus –Numerical Rating Scale (P-NRS). At the beginning of treatment, patients rated their itchiness at more than 8.9 points on the P-NRS. However, after 16 weeks of treatment, itching significantly decreased to 0.2 points. These results exceed those reported in clinical trials. This suggests that upadacitinib has the potential to significantly improve the quality of life of AD patients by effectively managing pruritus [159].

### 3.3. Differences and Similarities in Pruritus in PSO and AD

#### Interleukin 13 Receptor Alpha 2 (IL-13Rα2)

Xiao et al. investigated regulatory circuits of IL-13Rα2 in patients with AD and PSO. Firstly, they found out that IL-13Rα2 mRNA levels, as well as IL-13 mRNA levels, were increased in AD skin samples when compared with psoriatic skin or healthy controls. These results imply that transcription of the molecules mentioned above is correlated with the activity of AD, but not so much with PSO. Secondly, with the use of calcium imaging and western blot analysis, Xiao et al. discovered that IL-13 treatment induced upregulation of IL-13Rα2 transcription levels in keratinocytes (Figure 2). Moreover, they established that IL-13Rα2 took part in the regulation of numerous inflammatory and itch mediator genes, such as EDNRA, NRP1, CCL20, CCL26, SERPIN family members, and TNF family members. This may suggest that IL-13 and its receptors are important for regulating skin barrier dysfunction and pruritus in AD. Finally, they performed an experiment on murine dorsal root ganglion neurons to assess the functional contribution of IL-13Rα2 to toll-like receptor 2 (TLR2)-regulated IL-13 responses in sensory neurons. This experiment was designed to investigate another mechanism underlying itch sensitization to IL-13. It appeared that TLR2 can upregulate peripheral IL-13Rα2 expression, which led to exaggeration of IL-13-mediated pruritus suggesting that peripheral TLR2-IL-13Rα2 signaling can cause itching sensation and enable neurogenic inflammation. These findings reveal a new mechanism of IL-13Rα2 which regulates itch and inflammation in AD, but not in PSO [160].

## 4. Discussion

In summary, pruritus is a common and concerning symptom in PSO and AD, driven by complex immunological and biochemical processes. The identification of common mechanisms underlying pruritus in these conditions opens new avenues for research and therapeutic interventions. Both PSO and AD involve complex multifactorial pathophysiologies underlying pruritus. These determinants include cytokines, interleukines, histamine, TRP channels, nerve growth factor, opioid receptors, DPP IV, and TSLP. It appears that the mechanisms underlying pruritus in PSO and AD may involve some overlapping mediators, but there clearly are some differences. The complex and interrelated nature of these mechanisms underscores the challenges of developing targeted therapies. Despite extensive clinical research, there is still no clear answer as to what triggers pruritus. However, it seems promising to focus further research on the genetic basis of pruritus and interleukins.

## 5. Conclusions

Each molecule involved in the pruritic pathway would merit a separate chapter or even an entire book, since, as we have shown, the pathomechanism of pruritus is an extremely complex and intricate process. Moreover, many of these signaling pathways are currently undergoing detailed analysis (hopefully with some clarification in the future) or are still unexplained. As a result, it is currently difficult to take an objective view of how far we have come in elucidating the pathogenesis of pruritus in the described diseases. Nevertheless, considerable progress has been made in recent years.

## Figures and Tables

**Figure 1 ijms-24-14734-f001:**
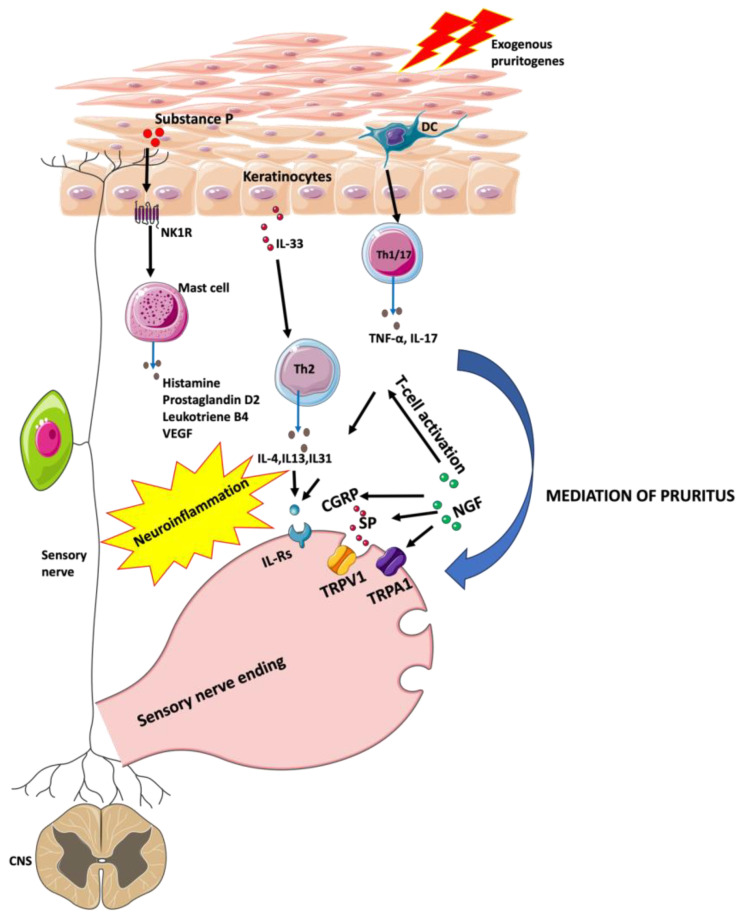
The scheme illustrates the simplified molecular pathways of itch in psoriasis. From left to right: Substance P acts via NK1R which leads to the degranulation of mast cells and the release of histamine, prostaglandin D2, leukotriene B4, VEGF, and TNF-α. Overexpression of IL-33 in keratinocytes induces the recruitment of Th2 cells which release cytokines such as IL-4, IL-13, and IL-31. Dendritic cells contribute to the upregulation of IL-17 and TNF-α production from activated T-cells. NGF activates the TRPV1. CGRP and SP are released upon activation of TRPV1. NGF up-regulates TRPA1. The Figure was partly generated using Servier Medical Art, provided by Servier, licensed under a Creative Commons Attribution 3.0 unported license.

**Figure 2 ijms-24-14734-f002:**
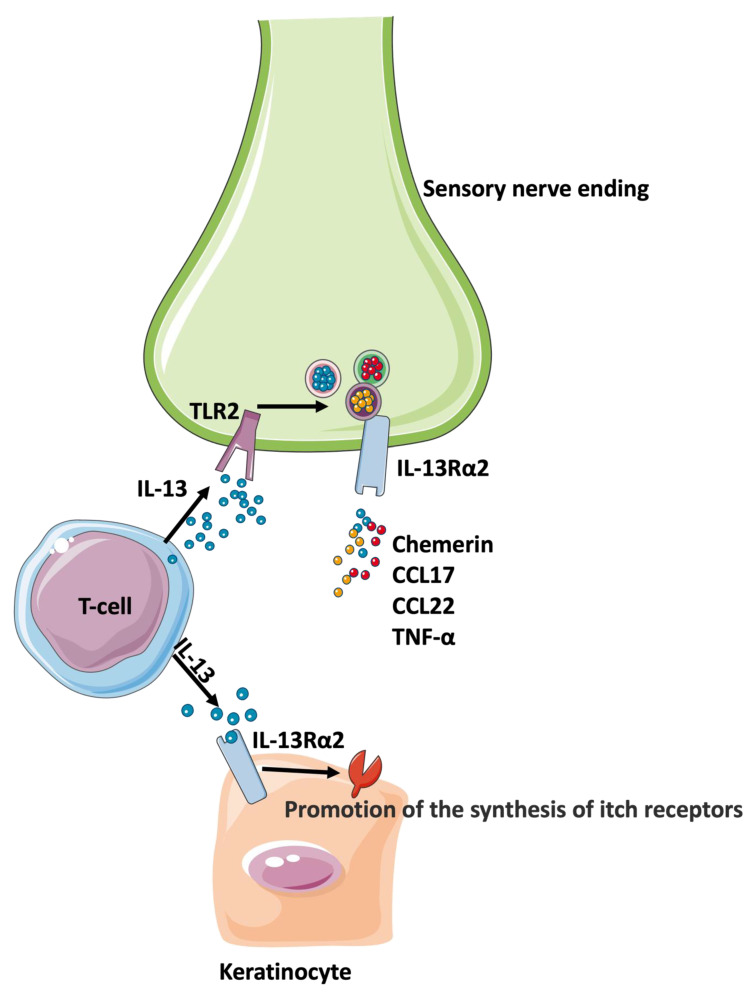
The scheme of a new mechanism of pruritus in AD. IL-13 mediates itch via TLR2-IL-13Rα2 signaling which promotes the release of chemerin, CCL17, CCL22, and TNF-α from sensory neurons. IL-13 upregulates IL-13Rα2 in keratinocytes which further promotes the synthesis of itch receptors. The Figure was partly generated using Servier Medical Art, provided by Servier, licensed under a Creative Commons Attribution 3.0 unported license.

## Data Availability

Data availability is not applicable to this article as no data sets were generated or analyzed during the current study.

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
