# Peer review of "Pathomechanism of Pruritus in Psoriasis and Atopic Dermatitis: Novel Approaches, Similarities and Differences"

_ijms, 2023, doi:10.3390/ijms241914734_

Round 1

Reviewer 1 Report

Thank you for the opportunity to review this manuscript by Kaczmarska and colleagues.  The goal of this review is to summarize the mechanisms of itch in psoriasis vs atopic dermatitis.  This is an important  area and there are many strengths to the present manuscript.  One in particular is that the authors importantly described studies taking advantage of psoriatics with/without pruritus.  My sense is that more detail is present regarding psoriasis over atopic dermatitis (AD).

However, I have several suggestions for improvement.

First, the authors missed an important opportunity to detail the sensitivity of AD patients to sweat as a pruritic stimulus--recently reviewed in https://doi.org/10.1111/exd.13981

I am surprised more details regarding Mrgpr receptors are not provided.

Finally, under the microbiome heading, the authors might wish to point out that pharmacologic levels of the TLR2 ligand lipoteichoic acid have been measured in infected AD skin lesions in children.  https://doi.org/10.1016%2Fj.jaci.2009.09.052

Author Response

Dear Sir or Madam, thank you very much for the review out manuscript entitled:
„Pathomechanism of pruritus in psoriasis and atopic dermatitis: novel approaches,
similarities and differences.”
In response to your comment, we would like to thank you for appreciating our
manuscript.
Ad 1 In verse 1231 we mentioned that external factors as well as sweat external
factors can contribute to the onset or severity of itching. We have also included the
work you suggested.
Ad 2 We have added an entire subsection 3.2.6. on Mas-related G-protein coupled
receptors (MRGPR) to the publication.
Ad3 In verses 1399-1407 we have marked the importance of TLR2 based on the
suggested article.
We do honestly hope that it will satisfy you and improve the quality of our work.
Once again we are very grateful for your review and remain open if you have any
other remarks or suggestions that will make our work merit publication in
“International Journal of Molecular Sciences”.

Reviewer 2 Report

The article is interesting and well written however it needs some major revisions

1)the authors talk about itching in these two conditions but there are experiences in the real life literature that need to be cited

- DOI: 10.1111/jdv.18137

- DOI: 10.1007/s00403-019-01998-7

2) the materials and methods section needs to be more detailed as we talk about a review

3)there are several typos and English errors so I recommend a total revision 

4) better to add discussion paragraph and non-summary conclusion, it definitely needs to be deepened this aspect

5) introduction should be improved with more detailed references and topics related to disease treatments

Minor editing of English language required

Author Response

Dear Sir or Madam, thank you very much for reviewing our manuscript entitled:
„Pathomechanism of pruritus in psoriasis and atopic dermatitis: novel approaches, similarities and differences.”
In response to your comment, we would like to thank you for appreciating our
manuscript.
Ad 1 The results of the suggested studies are presented in the paper and described:
the study by Napolitano et al. (verses 1461-1468, reference 160) and Hawro et al. in (verses 400-407, reference 61).
Ad 2 The materials and methods section has been improved and more details are included (verses 244-255)
Ad 3 The manuscript has been corrected by the authors for typos and English errors.
Ad 4 Added discussion paragraph and summary(verses 1488-1502)
Ad 5 The introduction has been improved with more detailed treatment reports citing clinical trials of drugs used to treat atopic dermatitis and psoriasis ( verses 39-104)
We do honestly hope that it will satisfy you and improve the quality of our work.
Once again we are very grateful for your review and remain open if you have any
other remarks or suggestions that will make our work merit publication in
“International Journal of Molecular Sciences”.

Reviewer 3 Report

In this review, Kaczmarska et al summarized current knowledge about the pathogenesis of pruritus in psoriasis and atopic dermatitis. They have concentrated on some reports which they found crucial in the understanding of pruritus. They concluded that it is currently difficult to take an objective view of how far they have come in elucidating the pathogenesis of pruritus in the described diseases. It is a comprehensive summary of the findings on itching to date, which will be useful to many readers. It is not complete, as research is still ongoing, but it is organized in a certain degree of clarity. There are some minor comments as follows.

minor concerns)

1) In line 126, you wrote "CCL/17TARC levels could be used as a biomarker of pruritus severity," but Please correct to "CCL17/TARC."

2) From line 331, "3.2. Pruritus in atopic dermatitis" seems to be in bold.

Also, there is no entry for 3.2.1. Please correct the numbering or add an item, as appropriate.

Author Response

Dear Sir or Madam, thank you very much for the review out manuscript entitled:
„Pathomechanism of pruritus in psoriasis and atopic dermatitis: novel approaches,
similarities and differences.”
In response to your comment, we would like to thank you for appreciating our
manuscript.
Ad 1 We corrected mistake in line 126 from CCL/17TARC to CCL17/TARC.
Ad 2 The style of headings and their numbering has been changed according to the
MDPI guidelines described in the Layout Style Guide.
We do honestly hope that it will satisfy you and improve the quality of our work.
Once again we are very grateful for your review and remain open if you have any
other remarks or suggestions that will make our work merit publication in
“International Journal of Molecular Sciences”.

Round 2

Reviewer 1 Report

Thank you for the opportunity to re-review this manuscript.  The additions have resulted in a stronger work.  I have no further concerns.

Reviewer 2 Report

Article is suitable for publication